# Structure of a cyanobacterial photosystem I tetramer revealed by cryo-electron microscopy

Koji Kato[1,8], Ryo Nagao[1,8], Tian-Yi Jiang[1,8], Yoshifumi Ueno[2], Makio Yokono [3], Siu Kit Chan[1], Mai Watanabe[4], Masahiko Ikeuchi[4], Jian-Ren Shen [1*], Seiji Akimoto [2*], Naoyuki Miyazaki[5,6*] & Fusamichi Akita [1,7*]

Photosystem I (PSI) functions to harvest light energy for conversion into chemical energy. The organisation of PSI is variable depending on the species of organism. Here we report the structure of a tetrameric PSI core isolated from a cyanobacterium, *Anabaena* sp. PCC 7120, analysed by single-particle cryo-electron microscopy (cryo-EM) at 3.3 Å resolution. The PSI tetramer has a C2 symmetry and is organised in a dimer of dimers form. The structure reveals interactions at the dimer-dimer interface and the existence of characteristic pigment orientations and inter-pigment distances within the dimer units that are important for unique excitation energy transfer. In particular, characteristic residues of PsaL are identified to be responsible for the formation of the tetramer. Time-resolved fluorescence analyses showed that the PSI tetramer has an enhanced excitation-energy quenching. These structural and spectroscopic findings provide insights into the physiological significance of the PSI tetramer and evolutionary changes of the PSI organisations.

[1] Research Institute for Interdisciplinary Science and Graduate School of Natural Science and Technology, Okayama University, Okayama 700-8530, Japan. [2] Graduate School of Science, Kobe University, Hyogo 657-8501, Japan. [3] Nippon Flour Mills Co., Ltd., Innovation Center, Kanagawa 243-0041, Japan. [4] Department of Life Sciences, Graduate School of Arts and Sciences, The University of Tokyo, Tokyo 153-8902, Japan. [5] Life Science Center for Survival Dynamics, Tsukuba Advanced Research Alliance (TARA), University of Tsukuba, Tsukuba, Japan. [6] Institute for Protein Research, Laboratory of Protein Synthesis and Expression, Osaka University, Osaka 565-0871, Japan. [7] Japan Science and Technology Agency, PRESTO, Saitama 332-0012, Japan. [8] These authors contributed equally: Koji Kato, Ryo Nagao, Tian-Yi Jiang. *email: shen@cc.okayama-u.ac.jp; akimoto@hawk.kobe-u.ac.jp; naomiyazaki@tara.tsukuba. ac.jp; fusamichi_a@okayama-u.ac.jp

Oxygenic photosynthesis performed by plants, algae and cyanobacteria plays a pivotal role in sustaining life on the earth by converting light energy into chemical energy and producing molecular oxygen from water[1]. The light energy conversion reactions are carried out by two multi-subunit pigment-protein complexes, photosystem I and photosystem II (PSI and PSII, respectively), embedded in the thylakoid membranes. It is generally accepted that PSII mainly forms a dimeric structure in all oxyphototrophs[2,3], whereas PSI exists in different oligomeric states in different organisms, i.e., monomeric[4–10] and trimeric PSI cores[11,12] in most eukaryotic and prokaryotic organisms, respectively. Recently, biochemical and electron microscopic analyses have shown that some cyanobacteria possess tetrameric PSI cores[13–16]. The structural differences and their functional implications between the different forms of PSI, however, are not clear.

The structures of cyanobacterial PSI core trimer and monomer have been solved by X-ray crystallography[11,12,17], and the structures of monomeric PSI core from eukaryotic algae and higher plants have also been solved by both X-ray crystallography and cryo-electron microscopy (cryo-EM) in the form of PSI-light harvesting complex I (LHCI) supercomplex[4–10]. These studies provide much information on the organisation of protein subunits, pigment arrangement and possible energy transfer pathways in the PSI core, as well as interactions among the PSI monomers in the cyanobacterial PSI trimer. The overall structure of PSI tetramer isolated from a cyanobacterium, *Chroococcidiopsis* sp. TS-821, was also reported by Semchonok et al. at ~12 Å resolution by single particle analysis with cryo-EM[18]. This resolution, however, was not enough to reveal the detailed arrangement of pigments and subunit interactions within the tetrameric PSI.

Here we solved the structure of the PSI tetramer isolated from a cyanobacterium, *Anabaena* sp. PCC 7120 (hereafter "*Anabaena*") by single particle cryo-EM analysis at 3.3 Å resolution, which reveals unique monomer-monomer interactions for the formation of the tetramer. We also compare excitation-energy dynamics of the PSI tetramer with those of PSI dimer and

monomer obtained from the same cyanobacterium by means of picosecond time-resolved fluorescence (TRF) spectroscopy. The combination of structural and spectroscopic studies provides important clues on the structural and functional differences between the trimeric and tetrameric PSI cores as well as the possible evolutionary adaptations of tetrameric PSI to specific light environment.

## Results

**Overall structure of the PSI tetramer**. The PSI core tetramers were prepared from *Anabaena*, and its biochemical and spectroscopic characterisations are summarised in Supplementary Fig. 1 together with PSI monomers and dimers obtained from the same cyanobacterium. It should be noted that this cyanobacterium appears to possess PSI dimer and monomer in addition to the PSI tetramer in vivo as suggested by native polyacrylamide gel electrophoresis (PAGE) analysis[13]. The native and SDS-PAGE analyses showed apparent differences in the overall molecular weight among the three PSI forms but little differences in their protein composition. The steady-state absorption and fluorescence properties are also very similar among the PSI monomer, dimer and tetramer, except a slight shift of the fluorescence emission to a longer wavelength in the PSI tetramer than those from the monomer and dimer (Supplementary Fig. 1d). The particles of PSI tetramer were visualised using a Titan Krios microscope equipped with a Volta phase-plate. The tetramer structure was reconstructed from 111,400 particles at 3.3 Å resolution based on the "gold standard" Fourier shell correlation (FSC) = 0.143 criterion (Fig. 1a, Supplementary Fig. 2, Supplementary Table 1) (Methods). The cryo-EM density map showed features of well-resolved side chains of most amino acid residues and prosthetic groups such as pigments and [4Fe–4S] clusters.

The overall atomic model of the PSI homo-tetramer was built based on the 3.3-Å cryo-EM map (Fig. 1b). The tetramer is organised in a dimer of dimers form with a C2 symmetry. Within a tetramer, the PSI monomers take two different, non-equivalent positions designated A and B, and their symmetrically related positions, A' and B', respectively (Fig. 1b). This is in agreement

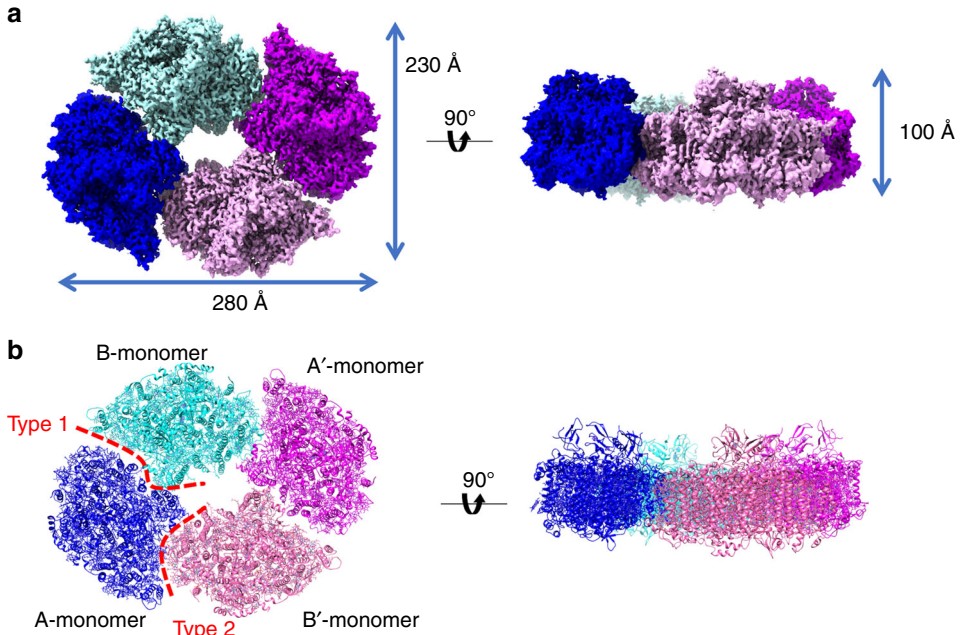

**Fig. 1** Overall structure of the PSI tetramer. **a** The 3D cryo-EM density map of the PSI tetramer viewed along the membrane normal from the stromal side (Left) and its side view (Right). **b** The structure of the PSI tetramer viewed along the membrane normal from the stromal side (Left) and its side view (Right)

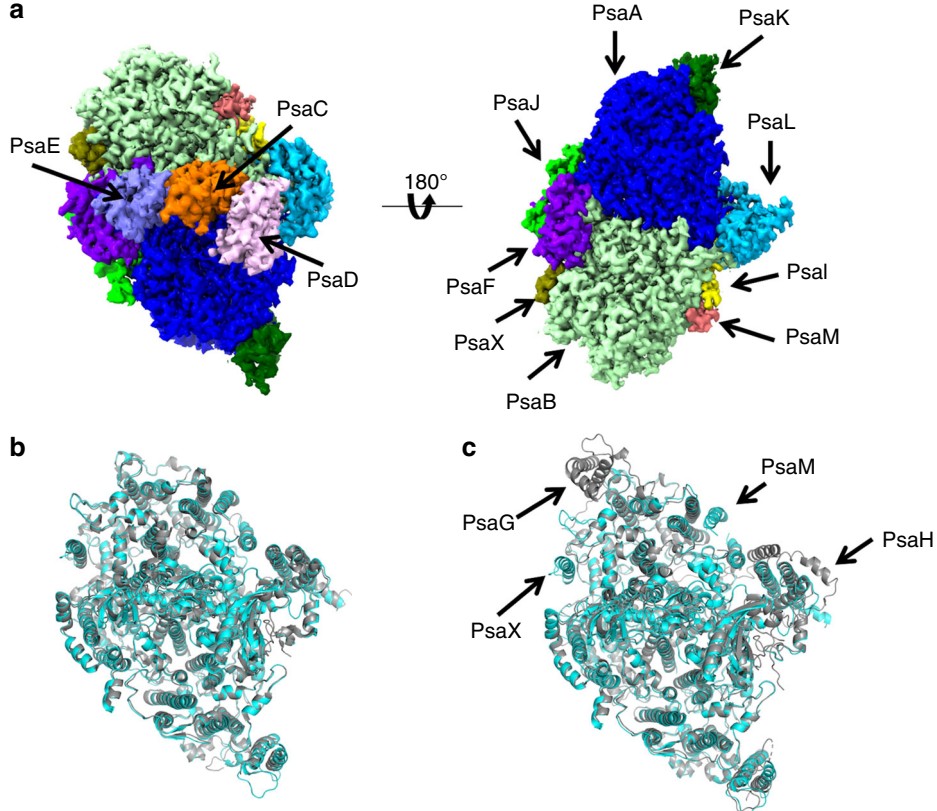

**Fig. 2** Structure of a PSI monomer. **a** The 3D map of a PSI monomer viewed along the membrane normal from the stromal (Left) and lumenal sides (Right). **b** Superposition of an *Anabaena* PSI monomer (cyan) with a *T. elongatus* PSI monomer (gray) viewed along the membrane normal from the stromal side. **c** Superposition of an *Anabaena* PSI monomer (cyan) with a *P. sativum* PSI monomer (gray) viewed along the membrane normal from the stromal side. The subunits specific to each organism are labeled

with the previous low-resolution cryo-EM structure of the PSI tetramer[18]. Thus, the neighbouring monomers have two different monomer interfaces, and we designate them here as Type 1 (between A-monomer and B-monomer) and Type 2 (between A-monomer and B′-monomer), respectively (Fig. 1b). In addition, we found that a part of the particles was in a dimeric form, and its structure was reconstructed at 4.0 Å resolution (Supplementary Fig. 3). The result showed that the dimer unit of PSI is formed by interactions between the A and B monomers. Thus, the tetramer is organised by A/B- and A′/B′-dimers. It should be noted that an extra density is found near PsaC and PsaE at the stromal side of PSI (Supplementary Figs. 2c and 3). However, we cannot identify the extra density because of the low resolution.

**Structure of a PSI monomer**. The structure of a PSI monomer unit within the tetramer is similar to that in the cyanobacterial trimer[11,12] and that in the higher plant PSI-LHCI super-complex[4,5]. Each monomer contains well-known nine membrane-spanning subunits (PsaA, PsaB, PsaF, PsaI, PsaJ, PsaK, PsaL, PsaM and PsaX) and three stromal subunits (PsaC, PsaD and PsaE) (Fig. 2 and Supplementary Fig. 4). In spite of the different oligomerization states, the positions and orientations of these subunits are well conserved in a monomeric unit among the cyanobacterial and plant PSI, except that the cyanobacterial PSI contains PsaM and PsaX, whereas the higher plant PSI lacks these two subunits but instead has the additional PsaG and PsaH subunits (Fig. 2b, c). The root mean square deviations (RMSD) not including the subunits that are present only in one species of PSI are 0.80 Å for 2,094 $C_\alpha$ atoms between the structures of *Anabaena* (present study) and *Thermosynechococcus elongatus*[11]

PSI monomers, and 0.97 Å for 1,988 $C_\alpha$ atoms between the *Anabaena* (present study) and *Pisum sativum*[4] PSI in their monomeric units (Fig. 2b, c). The cofactors identified in each monomer of the tetramers are summarised in Supplementary Table 2. There are 95 chlorophylls (Chl) *a*, 22 β-carotenes, 3 [4Fe-4S] clusters, 2 phylloquinones and 5 lipid molecules in a PSI monomer. The location of these molecules is largely similar to that in the cyanobacterial trimeric PSI structures[11,12].

**Interactions at the Type 1 interface**. The Type 1 interface refers to the interface between monomer A and monomer B within a dimer, and is formed by a number of interactions among the subunits PsaB (A), PsaL (A) and PsaL (B) (Figs. 1a, 3a). At the stromal side, PsaL (A) associates with PsaL (B) through hydrophobic interactions between F60/L64/A145/I148 from PsaL (A) and F60/L64/A145/I148 from PsaL (B) (Fig. 3b). At the lumenal side, PsaL (A) associates with PsaL (B) also through hydrophobic interactions between L84/L92/A93/L96 from PsaL (A) and F159/L166/I167/L170 from PsaL (B) (Fig. 3c). In addition, N140, N141 and S144 from PsaL (A) are in hydrogen bond distances with T58 and R61 from PsaL (B) (2.7–2.9 Å), which may further stabilise the dimer unit (Fig. 3d). On the other hand, PsaB (A) associates with PsaL (B) through hydrophobic interactions between F151/L155/F161 from PsaB (A) and L11/P12/P15 from PsaL (B), and Q158 of PsaB (A) is hydrogen-bonded to S13 of PsaL (B) at the stromal side (2.7 Å) (Fig. 3e). Furthermore, pigment–protein interactions are also involved in the dimer formation, which are found between S110 of PsaL (A) and Chl *a* (hereafter "CLA") 201 of PsaL (B) (2.7 Å), I148/I151 of PsaL (A) and β-carotene (BCR)−205 (3.9 Å) of

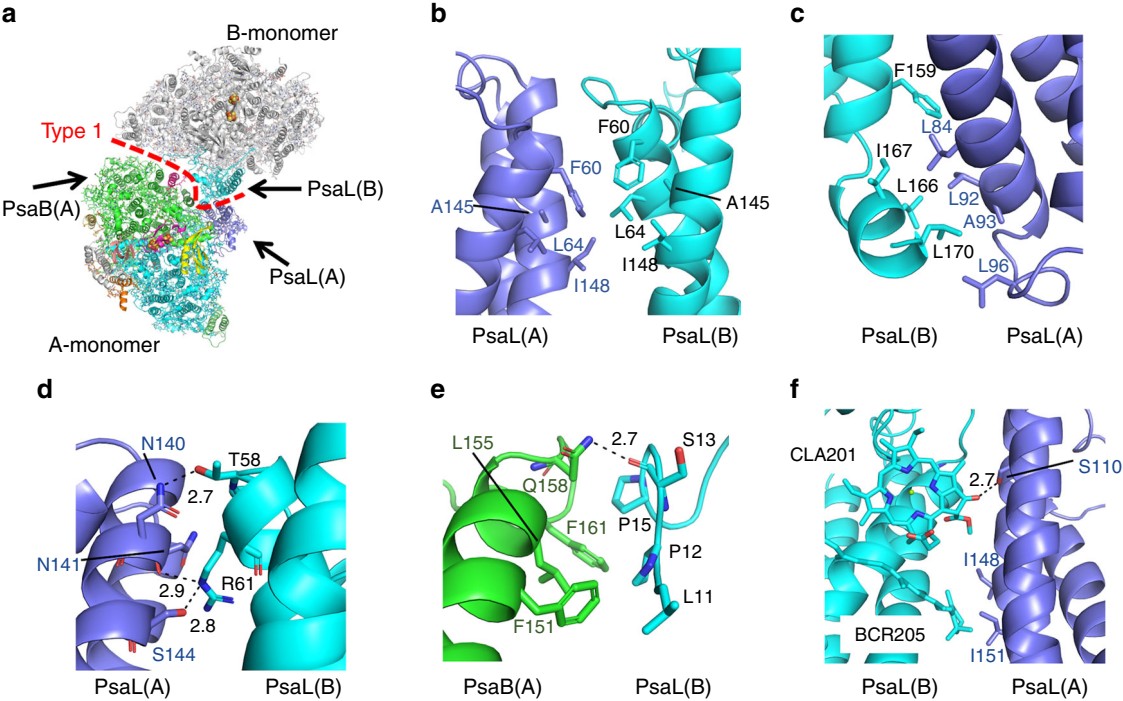

**Fig. 3** Interactions at the Type 1 interface. **a** Structure of a dimer unit (dimer A-B) from a PSI tetramer viewed along the membrane normal from the stromal side. **b**–**d** Interactions between PsaL (A) and PsaL (B). **e** Interactions between PsaB (A) and PsaL (B). **f** Pigment interactions between PsaL (A) and PsaL (B)

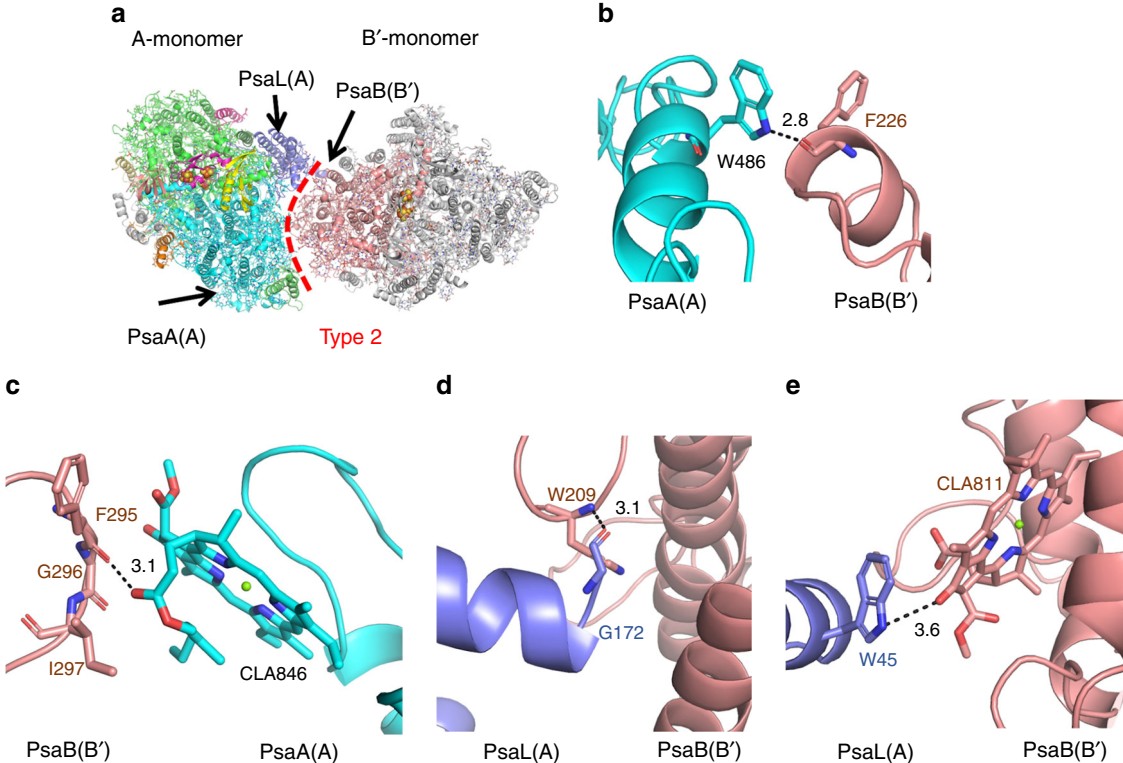

**Fig. 4** Interactions at the Type 2 interface. **a** Structure of a dimer unit (dimer A-B′) from a PSI tetramer viewed along the membrane normal from the stromal side. **b**, **c** Interactions between PsaA (A) and PsaB (B′). **d**, **e** Interactions between PsaL (A) and PsaB (B′)

PsaL (B) (Fig. 3f, some of the numbering of pigments and other cofactors discussed in the text are different from those registered in the PDB file; for a complete correspondence of the numbering of pigments and cofactors in the text with those in the PDB file, refer to Supplementary Tables 3).

**Interactions at the Type 2 interface**. The Type 2 interface refers to the interface between the two dimers within the PSI tetramer, and is formed between monomers A and B′ (Figs. 1b, 4a). These two monomers are connected mainly by hydrophilic interactions between PsaA (A)-PsaB (B′) and PsaL (A)-PsaB (B′). This Type 2

interface is unique for the PSI tetramer and not found in the PSI trimer. At the lumenal side, PsaA (A) interacts with PsaB (B′) through a hydrogen bond between W486 of PsaA (A) and the main-chain oxygen of F226 of PsaB (B′) (Fig. 4b). At the stromal side, an oxygen atom from the tail of CLA846 bound to PsaA (A) forms a hydrogen bond with the backbone carbonyl of F295 of PsaB (B′) (3.1 Å) (Fig. 4c). In addition, the C terminal helix of PsaL (A) is associated with PsaB in the B′-monomer at the lumenal side, and the main-chain oxygen of G172 from PsaL (A) is hydrogen-bonded with the main-chain nitrogen of W209 from PsaB (B′) (3.1 Å) (Fig. 4d). Furthermore, close contact is found between side chain nitrogen of W45 from PsaL (A) and C13 group carbonyl of CLA811 from PsaB (B′) (3.6 Å) at the stromal side (Fig. 4e).

**Structural basis for the oligomerisation of PSI**. The Type 1 interface in the PSI tetramer is similar to that of the monomer-monomer interface found in the PSI trimer, and most of the residues that contribute to the interactions are conserved across cyanobacteria. However, residues L11-P15, R61, N140 and S144 of PsaL exhibit high conservation among cyanobacteria having PSI tetramers, but they are replaced by relatively small residues in the species with trimeric PSI cores (Supplementary Fig. 5). To assess the effects of the characteristic amino acids on the oligomeric states of PSI, a monomeric unit of the tetrameric PSI cores was superposed on one of the monomers in the trimeric PSI (Supplementary Fig. 6). Surprisingly, we found a number of steric hindrances for the formation of the trimeric core using the monomeric PSI structure from the tetramer. These steric hindrances are mainly contributed by the N-terminal, middle and C terminal regions of PsaL, which include residues between L11-R16 in the N terminal region of PsaL (B) and PsaL-F151, Q158-F161 from PsaB (A), residues P55-R61 in the middle region of PsaL (B) and F60, T106-S110, K136-S144 of PsaL (A), and residues L170-V171 in the C terminal region of PsaL (B) and W92, P94 from PsaB (A) (Supplementary Fig. 6b, c, d). Multiple sequence alignment of different PsaL subunits from the trimeric PSI (*Synechocystis* sp. PCC 6803, *Thermosynechococcus elongatus* BP-1, *Arthrospira platensis* NIES-39, *Cyanothece* sp. ATCC 51142 and *Prochlorococcus marinus* SS120) and tetrameric PSI (*Anabaena* sp. PCC 7120, *Chroococcidiopsi* sp. TS-821, *Nostoc* sp. PCC 7524, *Calothrix* sp. PCC 7507 and *Anabaena variabilis* ATCC 29413) show that the bulky amino acids (T58, F60, R61, N140 and S144 of PsaL) are conserved only in the species forming the PSI tetramer (Supplementary Fig. 5)[16]. Therefore, these conserved amino acid residues of PsaL with large side chains appear to prevent formation of the trimeric PSI core.

Compared with the Type 1 interface, the Type 2 interface has a smaller number of protein-protein and pigment-protein interactions between the monomers, suggesting their weaker association. This also implies that the two dimers in the tetramer are relatively easy to be dissociated, thereby explaining the dimer of dimers organisation of the tetramer as well as the presence of dimers in the cryo-EM images of the tetramers. The main components of interactions at the Type 2 interface are hydrogen bond interactions among the main-chain atoms, and between main chain atoms and the Trp residues conserved in both tetramer and trimer-type cyanobacteria. These characteristics are common features between the PSI tetramer and trimer, as the structures of the main chains are essentially the same between the tetramer and trimer. In other words, the organisation of the PSI tetramer may be determined by the large side chain of PsaL at the Type 1 interface, that is, once a PSI dimer is formed by the specific interactions through the Type 1 interface, the two dimer units are assembled into a tetramer by the relatively weak and non-specific interactions observed at the Type 2 interface.

Based on these observations, we can draw a model for the assembly of the PSI cores. In the case that the Type 1 interface allows formation of the trimer, three PSI monomers are assembled by the relatively strong interactions through the Type 1 interface with each other, leading to the formation of a trimer. On the other hand, when the Type 1 interface does not allow formation of a trimer, only dimer is formed by the Type 1 interface, and further interactions between the two dimers are possible through the Type 2 interface, resulting in the formation of a tetramer. These structural findings, together with the sequence alignment of PsaL, suggest that PsaL with bulky amino acid residues is a key regulator for the formation of the PSI tetramer in cyanobacteria.

It is interesting to note that eukaryotic organisms, such as plants and algae, possess monomeric PSI cores[4–10]. This monomeric organisation is likely caused by the presence of the specific subunit PsaH in the eukaryotic PSI only, which is located near PasL (Fig. 2c). The close interactions of PsaH with PsaL probably prevent the formation of dimers, trimers or tetramers, and thus may allow a stable formation of the monomeric PSI cores in the eukaryotes[4–10,19,20]. Finally, the formation of the PSI monomer may facilitate the association of membrane-spanning LHCIs at the periphery of the PSI core together with the further acquisition of the PsaG subunit, forming the monomeric PSI-LHCI supercomplex[4–10]. In any event, the present results illustrated the important role of PsaL in the different patterns of oligomerizations of PSI during the evolutionary process.

**Excitation-energy dynamics in the PSI tetramer**. To examine the functions of the PSI tetramer, we analysed the excitation-energy dynamics of the PSI tetramer, dimer and monomer by picosecond TRF spectroscopy. The TRF spectra of the three PSI cores are largely identical, with a peak maximum at around 730 nm (Fig. 5a). There is, however, a slight decrease in the fluorescence intensity at around 715 nm in the spectrum of the PSI-tetramer compared with those of the monomer and dimer in the time range of 50 ps to 5.0 ns (Fig. 5a). The fluorescence decrease in the PSI tetramer is verified by the fluorescence decay curves monitored at 730 nm (Fig. 5b), reflecting a faster quenching of the excitation-energy in the PSI tetramer than that in the PSI monomer and dimer. This result suggests that the tetrameric organisation facilitated a faster energy quenching, through either energy trapping at the reaction center Chls or non-photochemical quenching by Chl–Car interactions, than that in the PSI monomer or dimer.

The slower decays of fluorescence in the PSI monomer and dimer than that in the PSI tetramer suggest that the dimer–dimer interactions, namely the Type II interface, within the tetramer should play a key role in the enhanced energy quenching. The increased quenching of fluorescence is caused by closer interactions of Chls which will lower the energy levels by Chl-Chl excitonic interactions. In the structure of the PSI tetramer, we found five clusters of triple Chls in a PSI monomer (Fig. 6), and these five clusters are conserved in the PSI cores irrespective of the tetramer and trimer[21] (Supplementary Fig. 7). As the triple Chls will significantly lower the Chl energy level, they are considered to contribute greatly to the energy quenching. In the Type 2 dimer-dimer interface unique to the PSI tetramer, cluster (Site) 2 in the B′ (B)-monomer closely interacts with cluster (Site) 5 in the A (A′)-monomer (Fig. 6c, f, g), which may contribute to the enhanced quenching observed in the PSI tetramer. These interactions may also facilitate inter-monomer energy transfer within the tetramer, conferring a physiological role of the PSI tetramer over the dimer and monomer. Two BCR molecules are located near the Site 2 and Site 5 with distances of 5.3–6.7 Å

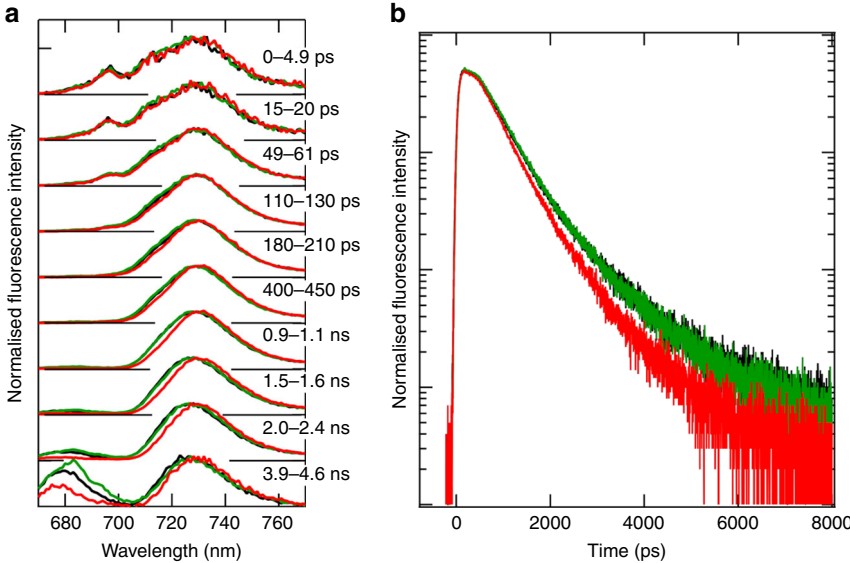

**Fig. 5** TRF analyses of the PSI cores. **a** TRF spectra measured at 77 K excited at 445 nm. The spectra of the PSI tetramer, dimer and monomer are depicted in red, green and black, respectively, and were normalised by the maximum intensity of each spectrum. **b** Normalised fluorescence decay curves at 77 K. The decay curves of the PSI tetramer, dimer and monomer are depicted in red, green and black, respectively, and were monitored at 730 nm

(Fig. 6g), which may mediate the energy quenching through these sites. Thus, the structural and spectroscopic findings strongly indicate that the Chl interactions between Site 2 and Site 5 in the dimer-dimer interface serve to quench the excitation energy through Chl to BCR energy transfer, and they are likely responsible for the longer wavelength shift of fluorescence maximum in the tetramer compared with that in the dimer and monomer (Supplementary Fig. 1). Once the Site 2/5 interactions in the dimer-dimer interface are broken, the fluorescence maximum appears to be shifted to a shorter wavelength with an increased peak intensity at around 682 nm (Supplementary Fig. 1d).

**Physiological significance of the tetrameric PSI core**. The TRF analyses showed a higher degree of excitation-energy quenching in the PSI tetramer than that in the PSI monomer and dimer from the same cyanobacterium (Fig. 5). This may have important physiological implications. Semchonok et al. reported that the PSI tetramer accumulated upon excess-light treatment (at a photosynthetic photon flux density (PPFD) of 800 µmol photons $m^{-2} s^{-1}$)[18], indicating that the PSI tetramer is formed in response to high light intensities. The higher quenching ability of the PSI tetramer found in the present study may be due to non-photochemical quenching rather than photochemical quenching, thereby suggesting that the tetrameric PSI core has a higher capacity of photoprotection, in consistent with its increased formation under a high light intensity[18]. The photoprotective response seems to occur by the excitation-energy transfer from the triply stacked Chls in the Site 2 and Site 5 through the BCRs located near these Chl triads, followed by rapid quenching of the excitation energy. By contrast, the PSI trimers from other cyanobacteria has only the Type 1 interface, which appears to have a slightly lower efficiency of non-photochemical quenching by the monomer-monomer interactions in the trimer cores.

## Discussion

The structure of PSI tetramer from *Anabaena* was solved at 3.3 Å resolution by cryo-EM, which showed an organisation of a dimer of dimers for the tetramer. Two types of interfaces were found to be required for the formation of the PSI tetramer. The Type 1

interface is similar to that of monomer-monomer interactions found in the PSI trimer from other species of cyanobacteria. In *Anabaena*, however, several residues of the PsaL subunit have larger side chains which hindered the formation of the PSI trimer. These residues are specific in the PsaL subunit from cyanobacteria that are known to form PSI tetramer and are conserved among them; therefore, the primary determinant for the formation of the PSI tetramer is the presence of these specific residues in PsaL. These large residues likely contribute to steric hindrances for the formation of PSI trimer, and they were changed to small residues from species possessing a PSI trimer.

The PSI tetramer was found to have a higher degree of excitation-energy quenching, likely non-photochemical quenching, providing an improved mechanism for photoprotection under excess light illumination. This is in agreement with the previous results showing that the amount of the PSI tetramer is increased under high light conditions[18]. Examination on the Chl interactions based on the structure obtained showed that a pair of Chl triads from two monomers of the tetramer located in the Type 2 interface interacts closely, which may be responsible for the enhanced energy quenching. This close interaction may also facilitate monomer–monomer energy transfer within the tetramer, providing an advantage for larger energy-harvesting and transfer capacities upon formation of the tetramer over dimer and monomer. The results obtained in the present study provide important clues on the assembly mechanisms and functional significance of the different PSI oligomeric states from different organisms, as well as their adaptations to the environment during the evolutionary process of PSI from prokaryotic to eukaryotic photosynthetic organisms.

## Methods

**Purification and characterisation of the PSI oligomers**. The cyanobacterium *Anabaena* sp. PCC 7120 was grown in BG11 medium supplemented with 10 mM HEPES-KOH (pH 8.0) at a PPFD of 30 µmol photons $m^{-2} s^{-1}$ at 30 °C with bubbling of air containing 3% (v/v) $CO_2$. Thylakoid membranes were prepared after disruption of the cells by agitation with glass beads on ice in the dark[22] and suspended in a buffer containing 0.2 M trehalose, 20 mM MES-NaOH (pH 6.5), 5 mM $CaCl_2$ and 10 mM $MgCl_2$ (buffer A). The thylakoids were solubilised with 1% (w/v) *n*-dodecyl-*β*-D-maltoside (*β*-DDM) at a Chl concentration of 1 mg mL$^{-1}$ for 30 min on ice in the dark with gentle stirring. After centrifugation at 20,000 × *g* for 20 min at 4 °C, the resultant supernatant was loaded onto a Q-sepharose anion-exchange column (2.5 cm of inner diameter and 10 cm of length)

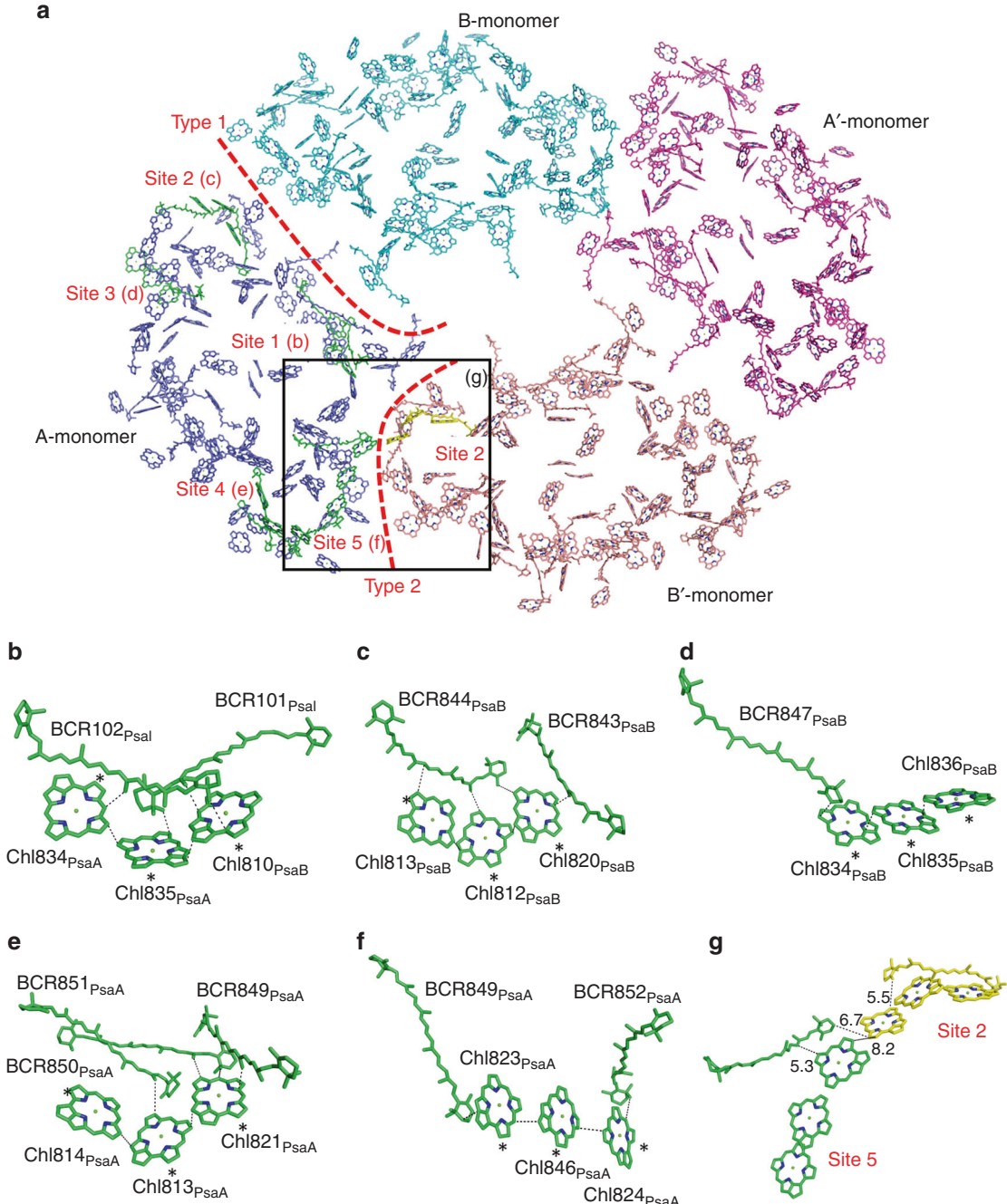

**Fig. 6** Arrangement of pigments in the PSI tetramer and their interactions. **a** Arrangement of the pigments in the PSI tetramer viewed along the membrane normal from the stromal side. Pigments of the A-monomer, B-monomer, A'-monomer and B'-monomer are coloured in blue, cyan, magenta and pink, respectively. Triply stacked Chls in the A-monomer, and a triple Chl cluster in the B'-monomer that is interacting with a triple Chl cluster in the A-monomer, are coloured in green and yellow, respectively. **b–f** Arrangement of the triply stacked Chls of the Site 1 (**b**), Site 2 (**c**), Site 3 (**d**), Site 4 (**e**) and Site 5 (**f**), in conjunction with their nearby carotenoids. Asterisks indicate the carbon atoms that are connected with a phytol tail. **g** The interfacial pigments between A-monomer (green) and B'-monomer (yellow)

equilibrated with buffer A containing 0.01% $\beta$-DDM (buffer B). The column was washed with a buffer containing 100 mM NaCl until the eluate became colourless, followed by elution with a 100–300 mM NaCl linear gradient at a flow rate of 2.0 mL min$^{-1}$ (total volume 400 mL). Three distinct peaks were eluted by the NaCl gradient in which, the first, second and third peaks were enriched in the PSI monomers, dimers and tetramers, respectively. Each PSI fraction was loaded onto a linear trehalose gradient of 10–40% (w/v) in a medium containing 20 mM MES-NaOH (pH 6.5), 5 mM CaCl$_2$, 10 mM MgCl$_2$ and 0.01% $\beta$-DDM. After centrifugation at 152,000 × g for 18 h at 4 °C (P40ST rotor; Hitachi, Japan), fractions of the PSI tetramer, dimer and monomer were obtained and then concentrated using a 100 kDa cut-off filter (Amicon Ultra; Millipore, USA) at 4000 × g. The concentrated PSI core complexes were stored in liquid nitrogen until use.

The three PSI cores were characterised biochemically and spectroscopically. Subunit composition of the PSI was analysed by a 16% SDS-PAGE containing 7.5 M urea according to the method reported previously[23] (Supplementary Fig. 1a). The samples (2 μg of Chl) were incubated for 10 min at 60 °C after the addition of 3% lithium lauryl sulfate and 75 mM dithiothreitol. A standard molecular weight marker (Precision Plus Protein Standards Dual Color, BioRad) was used. The subunit bands separated were assigned based on a previous study[14]. Clear native (CN)-PAGE was performed using a 3–8% polyacrylamide gel as described previously[24], and 2 μg Chl of each PSI sample was loaded in each lane (Supplementary Fig. 1b) together with a molecular marker (NativeMark Unstained Protein Standard; Invitrogen). Absorption spectra of the PSI cores were measured at 77 K using a spectrometer equipped with an integrating sphere unit (V-650/

ISVC-747, JASCO, Japan)[25] (Supplementary Fig. 1c), and their steady-state fluorescence spectra were recorded at 77 K using a spectrofluorometer equipped with an integrating sphere unit (FP-6600/ILFC-543L, JASCO)[26] (Supplementary Fig. 1d). The TRF spectra were recorded at 77 K by a time-correlated single-photon counting system with a wavelength interval of 1 nm/channel and a time interval of 2.44 ps/channel[27]. A picosecond pulse diode laser (PiL044X; Advanced Laser Diode Systems, Germany) operated at 445 nm with a repetition rate of 3 MHz was used as the excitation source. Pigment compositions were analysed using a Shimadzu HPLC comprising LC-20AD pumps and a SPD-M20A detector equipped with a reversed phase Inertsil C8 column (GL Sciences, Japan)[28], and the elution profile was monitored at 440 nm (Supplementary Fig. 1e).

**Cryo-EM data collection.** For cryo-EM experiments, 3-µL aliquots of the tetrameric PSI supercomplexes (14 µg Chl mL$^{-1}$) in a buffer containing 20 mM MES-NaOH (pH 6.5), 5 mM MgCl$_2$, 5 mM CaCl$_2$ and 0.04% (w/v) β-DDM were applied to Quantifoil R2/1, Mo 300 mesh grids covered with 5–10 nm amorphous carbon film. The grids were incubated for 30 s and then washed once with a washing buffer without treharose [20 mM HEPES (pH 7.0) and 0.04% (w/v) β-DDM] in a chamber of an FEI Vitrobot Mark IV at 4 °C and 100% humidity. The washed grids were immediately plunged into liquid ethane cooled by liquid nitrogen and then transferred into a cryo-electron microscope (Titan Krios, Thermo Fischer Scientific) equipped with a field emission gun, a Cs corrector (CEOS GmbH), a Volta phase plate, and a direct electron detection camera (Falcon 3EC, Thermo Fischer Scientific). The microscope was operated at 300 kV and a nominal magnification of 59,000. Approximately 5,000 Volta phase contrast movies were recorded using the Falcon 3EC detector in a linear mode with a pixel size of 1.12 Å and a total electron dose of 40 electrons Å$^{-2}$. Each exposure of 2.0 s was dose-fractionated into 26 movie frames. The nominal defocus range was −0.6 to −0.8 µm.

**Cryo-EM image processing.** The movie frames obtained were aligned and summed using the MotionCor2 software[29] to obtain a dose weighted image. Estimation of the contrast transfer function (CTF) including additional phase shift by the Volta phase plate was performed using the GCTF program[30]. All of the following processes were performed using the program RELION[31]. For structural analyses of the PSI tetramer, 1,853,015 particles were automatically picked from 5,060 micrographs and then used for reference-free 2D classification. For structural analysis of the PSI tetramer, in total, 405,731 particles were selected from good 2D classes and subsequently subjected to 3D classification without imposing any symmetry. The initial model for the first 3D classification was generated de novo from 2D classification. As shown in the Supplementary Fig. 8d, the PSI tetramer structure was reconstructed from 111,400 particles at an overall resolution of 3.3 Å.

Some particles from the cryo-EM images were found to be in a dimeric form, so they were subjected to 2D classifications separately. In total 164,700 particles of such PSI dimers were selected from good 2D classes and subsequently subjected to 3D classification without imposing any symmetry. The initial model for the first 3D classification was generated de novo from the 2D classification, and the final PSI dimer structure was reconstructed from 117,137 particles at an overall resolution of 4.0 Å (Supplementary Fig. 8d). All of the resolution was estimated by the golden FSC curve with a cut-off value of 0.143 (Supplementary Fig. 2)[32]. Local resolutions were estimated using Resmap[33] (Supplementary Fig. 2).

**Model building and refinement.** The 3.3-Å cryo-EM map was used for model building of the PSI tetramer. First, the crystal structure of *T. elongatus* PSI (TePSI, PDB codes: 1JB0) was manually fitted into the 3.3-Å cryo-EM map using UCSF Chimera[34], and then inspected and adjusted individually with Coot[35]. The amino acid sequences of the *T. elongatus* PSI structural model was then mutated to its counterparts from *Anabaena* sp. PCC 7120. The complete PSI tetramer structure was then refined with phenix.real_space_refine[36] with geometric restraints for the protein–cofactor coordination. For structural analysis of the PSI dimer, PSI A/B-dimer within the tetramer was manually fitted into the 4.0-Å cryo-EM map using UCSF Chimera. The final model was further validated with MolProbity[37] and EMringer[38]. The structure of the PSI dimer was identical with the PSI A/B-dimer in the tetramer. The statistics for all data collection and structure refinement are summarised in Supplementary Table 1.

**Reporting summary.** Further information on research design is available in the Nature Research Reporting Summary linked to this article.

## Data availability
The cryo-EM maps for the reported structure of the PSI tetramer and the PSI dimer have been deposited in the Electron Microscopy Data Bank under the accession codes EMD-9807 and EMD-9877, respectively. The atomic coordinates of the PSI tetramer have been deposited in the Protein Data Bank under the accession code 6JEO. Other data are available from the corresponding authors upon reasonable request.

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

## Acknowledgements

This work was supported by the Platform Project for Supporting Drug Discovery and Life Science Research (Basis for Supporting Innovative Drug Discovery and Life Science Research (BINDS) from AMED under Grant Number JP18am0101072j002 (to N.M.), and JSPS KAKENHI Nos. JP19H04726 (to R.N.), JP17H0643410 (to J.-R.S.) and JP16H06553 (to S.A.).

## Author contributions

F.A., N.M. and J-R.S. conceived the project; R.N., T-Y.J. and S.K.C. purified the PSI cores, and characterised the biochemical features; R.N., Y.U., M.Y. and S.A. measured the steady-state and time-resolved spectroscopies and analysed the data; M.W. and M.I. provided the culture and amino acid sequence information of *Anabaena* sp. PCC 7120; F.A., and N.M. collected cryo-EM images; K.K., R.N. and N.M. processed the EM data; K.K. built the structure model, refined the final models and analysed the structure; and K.K., R.N., F.A., S.A., N.M. and J-R.S. wrote the paper, and all of the authors joined the discussion of the results.

## Competing interests

The authors declare no competing interests.
