## [Peer Review File · Nature Communications]

Reviewers' comments:

Reviewer #2 (Remarks to the Author):

Overview (repeated from first revision): Kato et al. describe the 3.3-angstrom cryo-electron microscopy structure of tetrameric photosystem I isolated from the cyanobacterium *Anabaena* sp. PCC 7120. The structural analysis identifies the overall "dimer of dimers" structure and provides insight into the two main monomer-monomer interactions, "Type 1", and "Type 2". The first, "Type 1" interaction is essentially the same as is observed in trimeric photosystem I. The second, "Type 2", is unique to the tetrameric form of photosystem I and importantly facilitates the energetic coupling of pigments involved in the dissipation of excess excitation energy that occurs under high-light conditions (this hypothesis is supported by time-resolved fluorescence results). Various molecular interactions are identified within the structure that provide insight into the diversity of energy dissipation mechanisms in phototrophy.

General: The manuscript is much improved from the first set of revisions. Pending the revisions listed below, this reviewer believes the manuscript is in good shape for publication.

Major comments:

- 1) Can the authors comment on why there appears to be so much extra density in the 4-angstrom map (Supplementary Fig. 3) that is not modelled-into? Some may be due to detergent micelle but there also seems to be excess density near the stromal subunits of the A-monomer.
- 2) From previous review: The PDB validation report still does not reflect the CL0 ligand, although the authors stated they made this change in the structure.

Minor comments:

- 1) Line 66: This reviewer suggests against the use of "near-atomic resolution" as this term seems like an exaggeration for 3.3-angstrom resolution, especially where some local resolution extends to ~5-angstrom resolution. It may be better to simply state that the global resolution is 3.3 angstroms. An interesting discussion regarding the term "atomic-resolution" and "near-atomic resolution" can be found here: Wlodawer and Dauter, 2017, "Atomic resolution: a badly abused term in structural biology", *Acta Cryst D*.
- 2) Line 73: What is meant by "advances"? Oligomeric states of PSI are a result of evolutionary adaptation to environmental niches (i.e. high light). This reviewer cautions against implying that one may be better (or more advanced) than another unless specifying an environmental condition.
- 3) Supplementary Fig. 1 legend: For C, please list the wavelength that was used for normalization rather than the statement "Qy" peak.
- 4) Line 86: Typo - change "slightly" to "slight".
- 5) Line 124: Type - "structure" should be plural, "structures".
- 6) Supplementary Fig. 5: Sometimes genus is abbreviated, sometimes it is not. Sometimes genus/species is italicized when they should all be. This should be corrected.
- 7) Supplementary Fig. 5: It does not list how the sequence alignment was made. If it was Clustal Omega as it appears to be, this reviewer suggests placing "Clustal Omega" in parentheses.
- 8) Line 213: Typo - "events" should be "event".
- 9) Line 262: Typo - "is" can be deleted.
- 10) Line 301: "Hepes" should be capitalized, "HEPES".
- 11) Lines 304 and 316: "Mes" should be capitalized, "MES".
- 12) Line 349: This reviewer suggests the authors remove, "state-of-the-art".
- 13) Supplementary Table 1: Typo - "Pixcel" should be "Pixel".

Responses to the reviewer's comments:

Reviewer #2:

Comment 1:

Remarks to the Author:

Overview (repeated from first revision): Kato et al. describe the 3.3-angstrom cryo-electron microscopy structure of tetrameric photosystem I isolated from the cyanobacterium *Anabaena* sp. PCC 7120. The structural analysis identifies the overall "dimer of dimers" structure and provides insight into the two main monomer-monomer interactions, "Type 1", and "Type 2". The first, "Type 1" interaction is essentially the same as is observed in trimeric photosystem I. The second, "Type 2", is unique to the tetrameric form of photosystem I and importantly facilitates the energetic coupling of pigments involved in the dissipation of excess excitation energy that occurs under high-light conditions (this hypothesis is supported by time-resolved fluorescence results). Various molecular interactions are identified within the structure that provide insight into the diversity of energy dissipation mechanisms in phototrophy.

General: The manuscript is much improved from the first set of revisions. Pending the revisions listed below, this reviewer believes the manuscript is in good shape for publication.

Author reply 1:

First of all, we thank the reviewer for his/her positive and valuable comments to improve our manuscript. According to the reviewer's comment, we modified the manuscript as follows.

Major comments:

Comment 2:

1) Can the authors comment on why there appears to be so much extra density in the 4-angstrom map (Supplementary Fig. 3) that is not modelled-into? Some may be due to

detergent micelle but there also seems to be excess density near the stromal subunits of the A-monomer.

Author reply 2:

The extra density pointed out by the reviewer is found in both PSI tetramer and PSI dimer (Supplementary Figs. 2c and 3). We could not build any models in this extra density because of their featureless nature, presumably due to the limited resolution. This extra density might be due to detergent micelle or some protein aggregation in the grid preparation process. In the revised manuscript, we added the sentence “It should be noted that an extra density is found near PsaC and PsaE on the stromal side of PSI (Supplementary Figs. 2c and 3). However, we cannot identify the extra density because of the low resolution.” to the last of section “Overall structure of a PSI tetramer” (line 110).

Comment 3:

2) From previous review: The PDB validation report still does not reflect the CL0 ligand, although the authors stated they made this change in the structure.

Author reply 3:

We sincerely apologize that the PDB validation report was not updated. Now, we uploaded the new validation report in which the corresponding ligand was changed to “CL0”.

Minor comments:

Comment 4:

1) Line 66: This reviewer suggests against the use of “near-atomic resolution” as this term seems like an exaggeration for 3.3-angstrom resolution, especially where some local resolution extends to ~5-angstrom resolution. It may be better to simply state that the global resolution is 3.3 angstroms. An interesting discussion regarding the term “atomic-resolution” and “near-atomic resolution” can be found here: Wlodawer and Dauter, 2017, “Atomic resolution: a badly abused term in structural biology”, Acta Cryst D.

Author reply 4:

Thanks for this important comment. According to the reviewer's comments, we revised "near-atomic resolution" to "3.3-Å resolution" on lines 33 and 67.

Comment 5:

2) Line 73: What is meant by "advances"? Oligomeric states of PSI are a result of evolutionary adaptation to environmental niches (i.e. high light). This reviewer cautions against implying that one may be better (or more advanced) than another unless specifying an environmental condition.

Author reply 5:

According to the reviewer's comments, we modified the sentence on line 72 to following sentence.

"The combination of structural and spectroscopic studies provides important clues on the structural and functional differences between the trimeric and tetrameric PSI cores as well as the possible evolutionary adaptations of tetrameric PSI to specific light environment."

Comment 6:

3) Supplementary Fig. 1 legend: For C, please list the wavelength that was used for normalization rather than the statement "Qy" peak.

Author reply 6:

According to the reviewer's comments, we added the wavelength "at 677 nm" into the Supplementary Fig. 1 legend.

Comment 7:

4) Line 86: Typo – change "slightly" to "slight".

Author reply 7:

We modified it; thank you.

Comment 8:

5) Line 124: Typo - “structure” should be plural, “structures”.

Author reply 8:

We modified it as suggested.

Comment 9:

6) Supplementary Fig. 5: Sometimes genus is abbreviated, sometimes it is not. Sometimes genus/species is italicized when they should all be. This should be corrected.

Author reply 9:

According to the reviewer’s comments, we revised all genus/species to italicized ones and added the strain numbers in the Supplementary Fig. 5.

Comment 10:

7) Supplementary Fig. 5: It does not list how the sequence alignment was made. If it was Clustal Omega as it appears to be, this reviewer suggests placing “Clustal Omega” in parentheses.

Author reply 10:

The sequence alignment was made with “CLUSTALW”, and we added “CLUSTALW” in the Supplementary Fig. 5 legend.

Comment 11:

8) Line 213: Typo – “events” should be “event”.

Author reply 11:

We modified it as suggested.

Comment 12:

9) Line 262: Typo – “is” can be deleted.

Author reply 12:

We removed “is” as suggested by the reviewer.

Comment 13:

10) Line 301: “Hepes” should be capitalized, “HEPES”.

Author reply 13:

We modified it as suggested.

Comment 14:

11) Lines 304 and 316: “Mes” should be capitalized, “MES”.

Author reply 14:

We modified it as suggested.

Comment 15:

12) Line 349: This reviewer suggests the authors remove, “state-of-the-art”.

Author reply 15:

We removed “state-of-the-art” as suggested by the reviewer.

Comment 16:

13) Supplementary Table 1: Typo – “Pixcel” should be “Pixel”.

Author reply 16:

We modified it as suggested.